# Brain Microvascular Pericytes—More than Bystanders in Breast Cancer Brain Metastasis

**DOI:** 10.3390/cells11081263

**Published:** 2022-04-08

**Authors:** Danyyl Ippolitov, Leanne Arreza, Maliha Nuzhat Munir, Sabine Hombach-Klonisch

**Affiliations:** 1Department of Human Anatomy and Cell Science, University of Manitoba, Winnipeg, MB R3E 0J9, Canada; ippolitd@myumanitoba.ca (D.I.); arrezljf@myumanitoba.ca (L.A.); munirmaliha@gmail.com (M.N.M.); 2Department of Pathology, University of Manitoba, Winnipeg, MB R3E 0Z2, Canada

**Keywords:** brain, pericytes, NVU, pericyte markers, breast-to-brain metastasis, pericyte study models

## Abstract

Brain tissue contains the highest number of perivascular pericytes compared to other organs. Pericytes are known to regulate brain perfusion and to play an important role within the neurovascular unit (NVU). The high phenotypic and functional plasticity of pericytes make this cell type a prime candidate to aid physiological adaptations but also propose pericytes as important modulators in diverse pathologies in the brain. This review highlights known phenotypes of pericytes in the brain, discusses the diverse markers for brain pericytes, and reviews current in vitro and in vivo experimental models to study pericyte function. Our current knowledge of pericyte phenotypes as it relates to metastatic growth patterns in breast cancer brain metastasis is presented as an example for the crosstalk between pericytes, endothelial cells, and metastatic cells. Future challenges lie in establishing methods for real-time monitoring of pericyte crosstalk to understand causal events in the brain metastatic process.

## 1. Breast Cancer Brain Metastasis

Breast cancer (BC) remains the leading cause of cancer death and disability-adjusted life years in women [1], and approximately one in eight women will be diagnosed with breast cancer in their lifetime [2]. Despite significant improvements in diagnosis, treatment, and outcomes, about 6% of all BC cases are diagnosed at the metastatic stage with a poor 5-year survival rate of less than 27% [3]. BC metastasis to the central nervous system (CNS) constitutes a serious complication that is a strong clinical indicator of poor prognosis and can coincide with destructive neurologic complications [4]. CNS lesions comprise 13–30% of all BC metastases [5,6]. The number of patients with breast cancer brain metastases (BCBM) may be higher than anticipated due to asymptomatic cases of BCBM identified in screenings [7,8] and autopsy-based studies [9].

BC is a heterogeneous disease that can be categorized based on multiple morphological and molecular properties [10,11]. One of the most widely used molecular classifications divides BC into four different subtypes based on the presence/absence of estrogen receptor (ER), progesterone receptor (PR), and human epidermal growth factor receptor 2 (HER2). This includes luminal A (ER+ and/or PR+, and HER2-), luminal B (ER+ and/or PR+, and HER2+), HER2-enriched (ER- and PR-, and HER2+), and basal-like or triple-negative (ER-, PR-, and HER2-) BC [12]. BCBM are more likely to occur in women with HER2-enriched BC (30–55%) and triple-negative tumors (24–46%). [13,14]. In addition, patients with hormone receptor-negative BC were more likely to relapse in the brain in the first 5 years compared to hormone receptor+ BC tumors [15,16].

BCBM can occur in different CNS regions. Rostami et al. reported that 52.2% of BCBM were supratentorial and 24.1% infratentorial, and 14% of patients had metastases at both locations [17]. Magnetic resonance imaging data suggest that the cerebellum (33%), frontal lobe (26%), and brain stem (5%) are the most common BCBM locations, and multiple lesions were observed in 54.2% [17]. These data are partially supported by postmortem studies [18], where BCBM were most frequently noted in the cerebellum, occipital lobe, and basal ganglia. While the brain parenchyma is the preferred location for BCBM to form, the choroid plexus and leptomeninges can also be sites in 5% of women diagnosed with BCBM [19,20]. Together, these data emphasize a prominent role of the CNS environment for the successful establishment of BCBM lesions. However, little is known about the molecular mechanisms that regulate the formation of BC brain metastases and define the complex cellular microenvironment at BC brain metastatic sites.

## 2. Brain Colonization of Breast Cancer Cells

Brain metastatic colonization is a multistep process that starts at the primary tumor site and includes epithelial-to-mesenchymal transdifferentiation (EMT) to enable BC cells to detach from the primary BC tumor, penetrate extracellular matrices, and invade neighboring tissues. The invasion of blood and lymph vessels signals a major metastatic leap and provides a selective advantage for the evolution of a heterogeneous group of tumor cells that have developed strategies to survive as circulating tumor cells (CTC) in the blood and lymph and crucially have the ability to colonize distant organs [21]. The establishment of brain metastatic sites is particularly challenging for CTC and is known to take longer than the colonization of other distant sites such as the lungs or the bones. It requires CTC capable of transitioning through a tightly controlled blood–brain barrier (BBB) to gain access to the brain microenvironment with its unique extracellular matrix, metabolic, and immunological challenges. BC brain metastasizing cells were shown to reside in brain capillaries in close contact with endothelial cells for up to 7 days [22]. The successful completion of this treacherous journey results in the formation of micro-metastatic BCBM lesions that have the potential to progress to macroscopic secondary tumors [23,24].

## 3. Breast Cancer Brain Metastatic Cells and the Neurovascular Unit (NVU)

The neurovascular unit (NVU) is an anatomically complex functional brain microenvironment that the brain metastatic BC cells encounter first when entering the brain vasculature (Figure 1). This NVU is a unique vascular niche composed of luminal brain microvascular endothelial cells (BMEC) on a basal lamina (BL) richly surrounded with pericytes on its abluminal side. These vascular structures form intimate contacts with cellular extensions of brain resident cells, including glial cells of astrocytic and oligodendroglial origin, neurons, and microglial cells to create a functional integrated neurovascular barrier critically important for normal brain functions. The NVU shows region-specific anatomical specialization, likely reflecting adaptations to the requirements of specialized niches within the brain microenvironment [25]. NG2 and PDGFRβ reporter studies in mice suggest that pericytes are more abundant in cortical layer I than in layers II and III [26], but this was studied in a defined region of the sensorimotor cortex only and likely does not represent all CNS regions.

The NVU is the site of entry of BC cells for the establishment of BCBM in the brain parenchyma [27,28]. Activation of both astrocytes and microglia coincides with the arrival and luminal attachment of CTC, suggesting a role for the NVU in sensing the non-resident (cancer) cells and acting as a potential early warning system against metastatic brain colonization even prior to cancer cells crossing the BBB [22,29]. A complex and dynamic structure separating blood from brain tissue, the BBB consists of non-fenestrated BMEC connected by tight junctions and adherens junctions, supported by a basement membrane (BM) and surrounded by astrocytic end-feet and PCs [30]. At the same time, the BBB is not just a physical barrier, it also acts as a selective transport interface, secretory body, and metabolic barrier [31].

This review focuses on pericyte populations at the NVU, their molecular markers, and the roles of these brain pericytes during BC cell colonization of the brain.

## 4. Pericytes Provide Key Functional Support for the NVU

Pericytes were first identified in 1873 when Charles-Marie Benjamin Rouget described a population of contractile cells surrounding the endothelial cells (EC) of small blood vessels. Originally, these cells were called “Rouget cells”, but later Zimmermann termed these cells “pericytes” (PC), as this name best describes their location around capillary vessels [32]. The vasculature in the human CNS has the highest density of pericyte coverage, with a 1:1–3:1 EC-to-pericyte ratio [33]. CNS pericytes are derived primarily from the neural crest. The neuroectodermal origin of these cells was confirmed using Wnt-1 Cre [34] and Sox10-Cre [35] fate mapping mouse models. However, CNS pericytes may also originate from the mesoderm and bone marrow. CNS pericytes of neuroectodermal origin comprise the most prevalent population destined for the forebrain, while pericytes of mesodermal origin may be found in the brainstem, spinal cord, and mid-brain [36,37]. Brain pericytes are responsible for the integration of endothelial functions with glial/astrocyte functions at the NVU [38], the regulation of cerebral blood flow [39], angiogenesis [40], BBB formation and integrity [41], neuroinflammation [42], and stem cell activity [43].

Pericytes regulate the BBB by reducing the expression of endothelial genes involved with transendothelial permeability (e.g., plasmalemma vesicle-associated protein (PLVAP)) and promote astrocyte end-feet polarization [32,33,44]. Crosstalk of pericytes with EC during early vasculogenesis aids in vascular cell development and maturation. The anti-angiogenic potential of pericytes helps with the proper stabilization of blood vessels by reducing the proliferation and migration of endothelial cells [45]. Pericyte-deficient mice exhibit BBB breakdown, allowing perivascular IgG accumulation in the hippocampus and cortex in an age-dependent manner. In contrast to wild-type controls, 6–8-month-old and 14–16-month-old platelet-derived growth factor receptor-beta (PDGFR-β)-deficient mice showed an approximately 8–10-fold and 20–25-fold greater IgG perivascular accumulation in the hippocampus and cortex, respectively [46]. This points to an important function of the pericyte receptor PDGFRβ in the brain endothelial cell differentiation and function. As the breakdown of the BBB typically indicates disruption of the BBB tight junctions, PDGFR-β-deficient mice showed a progressive age-dependent reduction in the expression of key tight junction proteins ZO-1, occludin, and claudin-5 by 40–50% at 6–8 months and 80% at 14–16 months of age, respectively [46]. Mice deficient in PDGF-B and PDGFR-β receptors exhibit abnormal capillary shape and morphology. In PDGFR-β-deficient mice, this translates into a 25% increase in mean blood vessel diameter compared to wild-type mice, confirming that pericyte contractile capabilities regulate blood flow [47,48]. These findings have established pericytes as a critical cellular NVU component that cooperates with EC to facilitate BBB integrity in an age-dependent manner in the mouse brain.

Pericytes also have critical roles in tumor vasculature. In fibrosarcoma and osteosarcoma mouse models, pericytes were shown to display aberrant PDGF signaling and, similar to mice lacking PDGF-B and its receptor, this coincides with increased blood vessel diameters and reduced endothelial cell junctional circumference in the tumor vasculature [49,50]. Pericytes with abnormal morphology are loosely attached to the tumor vessels and show rapid turnover, thus rendering these vessels leaky [50,51]. Pericytes have stem cell ability and can differentiate into multiple types of mesenchymal precursor cells: fibroblasts, osteoblasts, chondroblasts, adipocytes, vascular smooth muscle cells, and skeletal muscle cell precursors. These cells express stem cell markers such as CD44, CD73, CD90, and CD105 [52]. Pericytes in the brain were also shown to have multilineage differentiation potential in vitro capable of differentiating into neural and vascular cells under hypoxic conditions [53]. This identifies the pericyte population as a source of regeneration and plasticity for functional gain of the NVU.

## 5. Pericyte Markers

Improper identification and frequent mix-ups with adjacent cell types, e.g., vascular smooth muscle cells (vSMCs) and juxtavascular microglia, have resulted in conflicting data being published [54]. Hence, the identification of specific pericytic markers is fundamental in better understanding the role of pericytes in normal and tumor environments, including BCBM. Pericytes can be definitively identified by electron microscopy (EM) features such as perivascular location with extensions of their slender ramified cytoplasmic processes along the capillary, an oval nuclear shape with high nucleus-to-cytoplasm ratio, and poorly developed cell organelles [55]. However, such morphological EM studies only provide very limited functional information and are not suitable for the selective isolation of defined populations of microvascular pericytes [56]. The diversity of pericytic phenotypes, the lack of a definitive pan-pericyte marker, and the differences in expression profiles of pericyte-associated genes in isolated pericytes versus in vivo samples are major challenges [45,57]. Currently, no single pericyte-specific marker is known, and all current markers used to identify pericytes are dynamic in their expression and may be up- or downregulated due to pathological or culture conditions [32]. The markers commonly used to identify brain pericytes include PDGFR-β [58], membrane alanyl aminopeptidase (CD13) [59], alpha-smooth muscle actin (αSMA) [60], neuron-glial antigen 2 (NG2) [61], melanoma cell adhesion molecule (MCAM or CD146) [62], and desmin [63]. The level of expression of a pericyte marker may also fluctuate as these multipotent cells possess self-renewing potential and display high phenotypic plasticity [64]. Hence, a state-of-the-art approach to identify pericytes in tissues must rely on a combination of tissue morphology, counter-labeling of EC and vSMC, and simultaneous staining for two or more pericyte markers [32]. Moreover, accurate identification of pericytes in vitro should also consider origin and co-culture conditions, as these cells tend to rapidly differentiate along multiple lineages depending on prevailing regulatory signals introduced with the specific culture conditions and the cellular microenvironment [65].

## 6. The NVU Contains Distinct Pericyte Populations

Several distinct types of pericytes have been described based upon the morphological appearance of their processes. This includes ensheathing, mesh, and thin-strand pericytes (Figure 2) [26]. The ensheathing pericytes cover mostly the pre-capillary portion of brain microvasculature, possess properties of both vSMCs and pericytes, and are considered a transitional form. The mesh and thin-strand pericytes of the capillary bed may be referred to as genuine capillary cells. Confocal imaging of thick coronal mouse brain sections has shown that only ensheathing pericytes demonstrate the level of αSMA expression comparable to vSMC, while αSMA expression was undetectable in both capillary mesh and thin-strand pericytes [57]. Similarly, prominent αSMA staining was exclusively found on relatively large precapillary arterioles [66]. Further evidence for the presence of distinct subtypes of pericytes was obtained during in vivo experiments in mice with a commercially available small molecule fluorescent dye, NeuroTrace 500/525, that labeled exclusively thin-strand non-contractile pericytes that lack α-SMA expression and possess long thin processes spanning multiple vessel branches [67]. By contrast, nearly all cultured pericytes started to express αSMA by day seven, which suggests differentiation of pericytes towards an ensheathing subtype in culture in the presence of a serum-containing medium [68]. In spite of the debates on the extent of differences between pericyte subpopulations [39,69], these in vitro data cast doubt on αSMA as a marker for the in vivo identification of true capillary pericytes.

The expression of CD146, also referred to as melanoma cell adhesion molecule (MCAM) or cell surface glycoprotein MUC18, was shown to be primarily confined to vSMC. High αSMA expression in CD146+ cells in vitro suggests that CD146 identifies a subpopulation of vSMC, but a role of CD146 as a pericyte marker is questionable [66]. These findings are supported by single-cell RNA sequencing data demonstrating that MCAM expression was approximately threefold higher in arterial and arteriolar vSMCs than in capillary pericytes [70]. Pericyte expression of NG2 also proved to be inconsistent. While one study suggested that 50–80% of isolated PDGFRβ+ cells were also NG2+ [71], another study demonstrated that prominent NG2 in situ staining was shown only in cases of focal cortical dysplasia but not in control tissues [72]. These data correlate with weak NG2 staining in human brain pericytes. NG2 is regarded as a plasticity marker with strongest expression at early stages of tissue development but declining later in ontogenesis [73,74]. NG2 might not be an appropriate target for the identification of quiescent human brain pericytes [75]. CD13 has been reported to have high specificity for capillary pericytes [26,38,59,66,76]. In addition, exclusive vascular staining of PDGFRβ may also serve as a suitable indicator for the presence of pericytes [38,66,77,78]. Collectively, the most reliable identification of brain pericytes under normal tissue conditions is based on a combination of parameters, including the double-positive staining for CD13 and PDGFRβ, the characterization of “bump-on-a-log” morphology with patterns for cell processes, and the selective counterstaining of vSMCs, EC, and astrocytes. Notably, the combined CD13/ PDGFRβ co-staining was shown to be most reliable for the identification and isolation of brain pericytes using FACS isolation, thus providing new exciting avenues for the study of brain pericytes [79]. The study of pericytes in different disease states, including neurodegenerative diseases and brain tumors, is anticipated to be critical in our understanding of the corresponding pathogenesis. Challenges to overcome include changes in protein marker expression within diseased tissues and cells, including tumor-associated pericytes acquiring an αSMA-positive phenotype likely reflecting tumor-induced angiogenesis and collagen synthesis [80].

## 7. The NVU in Breast-to-Brain Metastasis

Among the most common and adverse scenarios associated with BBB disruption are intracranial metastatic lesions [81]. The establishment of brain metastasis is a multistep process eventually leading to successful formation of brain macro-metastases (Figure 3A). The essential steps in this multistep metastatic process include the initial apposition and arrest of CTC at the endothelial luminal surface of blood vessels, early extravasation, perpetuated perivascular localization, vessel co-option, and angiogenic sprouting [82]. The presence of pre-existing blood vessels for vascular co-option and the close physical contact with the abluminal surface of the blood vessels are critical prerequisites for the survival of cancer cells in perivascular metastatic loci. The initial adaptation and survival are considered the most inefficient steps in the metastatic process and represent a stage of high vulnerability for these cancer cells towards the formation of micro-metastatic lesions [82]. Importantly, tumor cells have the ability to recruit pericytes to promote angiogenesis and, at the same time, displacing them from their initial vascular niche to enhance the leakiness of blood vessels [83]. The underlying mechanism is primarily dependent on paracrine PDGFRβ/ PDGF-BB signaling [84], while pro-angiogenic factors, such as hypoxia-induced factor 1α (HIF1α) and vascular endothelial growth factor (VEGF), determine both vessel co-option and angiogenesis [85,86] (Figure 3B). Enhanced stem cell properties and phenotypical plasticity of pericytes emerge as additional mechanisms by which pericytes may prepare the BC brain metastatic vascular niche to promote metastatic progression. The ability of pericytes (PC) to give rise to cancer-associated fibroblasts that promote tumor growth and dissemination was demonstrated in ovarian cancer [87].

Disruption of BBB integrity in BCBM arises from direct cell–cell interactions between BC cells and the NVU and the secretion of a broad range of cytokines/chemokines derived from tumor cells. In fact, CX3CL1 and CXCL13, also known as B cell-attracting chemokine (BCA-1), are notably elevated in patients with BCBM and associated with poor outcomes [88]. CX3CL1 can trigger CXCR1-expressing cancer cells to invade neighboring tissues [88], while CXCL13 may induce EMT of BMEC by binding to the receptors CXCR5 and CX3CR1 [89]. These observations were corroborated by the observation that CX3CL1 and CXCL13 containing serum obtained from BCBM patients significantly increased paracellular permeability of BMEC monolayers [90]. In addition to vessel co-option and angiogenesis, tumor vascularization may be explained by vascular mimicry (VM) (Figure 3C). VM refers to the formation of fluid-conducting networks by non-endothelial cells and has been reported for melanomas, sarcomas, breast, ovary, lung, and prostate carcinomas, and glioblastoma [91,92,93,94]. Breast cancer stem-like cells of MDA-MB-231 and SK-3 lines can differentiate into cells with endothelial markers, morphology, and function. In this way and independent of EC, breast cancer cells and BCBM can assume dual functions by attracting adjacent normal cells and forming primitive blood-vessel-like structures themselves [95]. The role of pericytes in VM was described in primary and brain metastatic melanoma. VM+ tumors are characterized by high PDGF-B secretion and a higher number of PDGFRβ+ pericytes assisting in stabilizing the vascular networks formed by VM+ cells [96,97]. The important supportive role of pericytes in maintaining brain endothelial barrier function becomes apparent from BBB culture models [98,99]. In vitro co-culture models demonstrated the ability of pericytes to suppress lung cancer cell migration through brain endothelial barriers [100]. In mouse models of breast and renal cell carcinoma as well as melanoma, complete loss of pericytes led to tumor hypoxia followed by EMT and increased metastasis [101]. These findings suggest that a loss of pericytes or their displacement from vascular structures can promote adverse local metabolic conditions that promote metastasis [83]. Importantly, tumor cells can transform the BBB into a blood–tumor barrier (BTB). The ability to induce BBB leakiness varies substantially among BC brain metastatic lesions [102]. While brain metastases of basal-like BC are reported to disrupt the BBB, this was less frequently observed in brain metastases of HER2-enriched breast cancers [103,104].

Vascular endothelial growth factor (VEGF) is implicated in the formation of BM and is frequently overexpressed and secreted by tumor cells in BM [105]. Pericytes express vascular endothelial growth factor receptor 1 (VEGFR1) [106] and stimulation of retinal pericytes with BC-derived VEGFR1 agonists, VEGF, or PlGF [107], resulting in pericyte vascular ablation, increased vascular leakage, and tissue edema [106]. This initiated a cascade of events leading to profound alterations in the behavior of angiogenic EC, which accumulated in thick protrusions but failed to form the normal number of vascular sprouts and branches [108].

## 8. Role of Pericytes in the Establishment of Metastatic Lesions

The initial steps of BC cell brain colonization have been described: (1) arrest of BC CTCs in brain capillaries, (2) BC cell passage through the BBB, (3) extravasation of BC cells from capillaries that are surrounded by PDGFRβ+ pericytes, and (4) initial growth of extravasated BC cells in the perivascular niche [22,82] (Figure 4). Upon successful extravasation and growth initiation, the fate of the metastatic cells is determined by the tissue milieu and cellular communication networks at the NVU niche [109].

The intravascular arrest of BC cells at the NVU includes cell–cell interactions of metastatic BC with microvascular EC, and this is sufficient to activate astrocytes. Upon extravasation, the formation of micro-metastatic lesions coincides with the appearance of activated astrocytes that accumulate both around and inside of the metastatic foci and form direct contacts with tumor cells [110]. The role of activated astrocytes in this brain metastatic process is largely unknown but may include supportive functions for nutrient transport, ion trafficking across the ECM, and neuronal signaling [22]. Co-culture experiments with astrocytes have demonstrated the ability of astrocyte-derived factors to induce a migratory response in BC cells [29]. Activated astrocytes can secrete potentially oncogenic factors such as interleukin 6 (IL-6), transforming growth factor beta (TGFb) and matrix metallopeptidase 9 (MMP-9) [111,112,113]. The astrocyte-derived secretion of the Erb-B2 receptor tyrosine kinase 3 (ErbB3, HER3) ligand neuregulin-1 (NRG-1) induces the proliferation, invasion, and BBB transmigration in Erb-B2 receptor tyrosine kinase 2 (ErbB2, HER2)-positive BC cells [114,115]. BCBM lesions were shown to have higher HER3 activity compared to the corresponding BC primary tumors [116], suggesting HER3 signaling may support metastatic growth. Activated microglia utilize Wnt signaling to promote the invasion and colonization potential of BC cells in brain metastatic lesions [117]. Unlike astrocyte activation, microglia activation was more variable and the differences in phagocytic activity and morphology observed may reflect the presence of different microglial subpopulations and/or different experimental conditions [109,118].

Despite the close proximity between pericytes, EC, and cancer cells during and after the trans-endothelial passage, the detailed role of brain pericytes in the process of BBB invasion and metastatic niche formation is largely unknown. BC dormancy in the brain perivascular niche was shown to be controlled by thrombospondin-1 secretion from well-differentiated EC [119]. Sprouting EC at developing branch points reduces thrombospondin-1 expression, which terminates BC cell quiescence and allows BCBM growth [119]. Although the role of pericytes in tumor dormancy at this early stage of BM is not known, tumor-cell-derived PDGF-BB leads to pericyte removal from the vessel and vessel sprouting [120]. Similarly, VEGF over-expression may alter pericyte activation and differentiation, resulting in increased CD31 expression, proliferation, and tumor angiogenesis of EC [105]. Several growth factors secreted by tumor cells, such as PDGF-B, VEGF-A, and TGFβ-1, may trigger the prevalence of a PDGFRβ+/desmin+ pericyte phenotype through the acquisition of αSMA, RGS5, NG2, and desmin immunopositivity in activated pericytes at BCBM sites [32]. Intriguing data from mouse models of BCBM indicate dynamic changes in pericyte subpopulations that coincide with altered BTB permeability and enhanced BCBM invasiveness. Mouse metastatic brain lesions arising from brain-trophic triple-negative MDA-MB-231 as well as HER2+ SUM190 and HER2+ JIMT-1 cell lines were highly permeable to Texas Red dextran, indicating a leaky BTB. These lesions showed an increase in PDGFRβ+/desmin+ and a decrease in PDGFRβ+/CD13+ perivascular cell populations presumed to reflect different pericyte subpopulations. In patients with brain metastasis, vessels of the unaffected brain were primarily surrounded by CD13+ pericytes, whereas desmin+ perivascular cells were primarily associated with brain metastatic lesions [105]. These observations are indicative of increased pericyte plasticity at BCBM lesions. Pericyte plasticity in the brain is not as well-documented as in other organs, but their stem cell capability in culture and their ability to differentiate into cells of vascular and neural lineages has been demonstrated in vitro [64]. PDGFRβ+ brain pericytes isolated from ischemic brain were shown to express the stem cell marker nestin and were capable to in vitro differentiate into vascular and myeloid lineages with phagocytic activity, suggesting microglial differentiation [121]. In human grade III and grade IV glioma with high PDGFβ expression levels, tumor microvessels were associated with increased microvascular α-SMA^+^ pericyte density and reduced CD31 staining, suggesting that pericytes take an active role in tumor microvessel formation [122]. Lineage tracing in vivo studies are needed to demonstrate the plasticity of pericytes in support of a metastasis-promoting function of brain pericytes.

Cancer cells and other cellular components of the tumor microenvironment respond to depleted oxygen levels with increased activity of hypoxia-inducible factors (HIFs) [123] and the activation of multiple HIF-1α target genes and downstream signaling cascades [124]. Importantly, HIF-1α stabilization leads to the increased expression and secretion of VEGF by astrocytes [125], which may promote the appearance of a PDGFRβ+//desmin+ pericyte vascular phenotype. Pericytes themselves respond to hypoxia with an upregulation of angiopoietin-1 (ANG-1) and rapid expression of VEGF [126,127]. Pericytes can also induce MMP expression in EC during hypoxia with downstream effects on extracellular matrix composition in the vicinity of the NVU niche. The abundance of perivascular pericytes in the CNS and their phenotypic and functional plasticity suggest that pericyte populations are rewarding targets for brain-metastasizing BC cells.

## 9. Pericytes and Endoplasmatic Reticulum (ER) Stress—An Emerging Science

Imbalances in the cellular homeostasis may generate cell stress from aberrant protein folding, which leads to ER stress. This causes the activation of the unfolded protein response (UPR) as an adaptive cellular response aimed at restoring homeostasis and cell survival [128]. ER stress may also affect pericyte viability and functions. Cultured retinal pericytes subjected to glucose deprivation or intermittent glucose reduction activate the ER transmembrane protein kinase PERK, which induces autophagy and the expression of VEGF-A and pro-inflammatory monocyte chemoattractant protein-1 (MCP1) [129,130]. Pericytes undergo regulated apoptosis resulting from ER stress induced by hypoglycemia or fluctuating glucose levels [129,131]. In glioblastoma brain tumor models, the activation of chaperone-mediated autophagy causes brain pericytes to release anti-inflammatory cytokines, such as TGF-β or IL-10, to block anti-tumor immune responses and instead promote brain tumor survival [132,133]. Recently, the transmission of ER stress from one cell to another cell was described as a process by which tumor cell-derived extracellular vesicles laden with functional proteins can initiate ER stress in other cells, such as macrophages, brain resident cells, and likely pericytes. This vesicular cargo can contain upstream regulators of UPR that activate UPR stress-related immunosuppressive responses to promote tumor cell survival, metastasis, and angiogenesis [130]. Intercellular transmissible ER stress (TERS) under metabolic stress was also shown to occur in hepatocytes through direct cell–cell contacts [134]. Information is currently lacking on adaptive ER stress responses between metastasizing cancer cells and perivascular pericytes, which influence the metastatic process. Emerging TERS research is likely to make ground-breaking discoveries on the multifaceted roles of pericytes in TERS at the NVU and in BCBM as well.

## 10. Pericytes and Metastatic Invasion Patterns

BM can be distinguished on the basis of their invasion patterns. Berghoff et al. proposed to categorize all metastatic brain lesions into three distinct types: well-demarcated, vascular co-option, and diffuse [135]. Another classification was later suggested by Teglasi et al., who distinguished pushing-type, papillary-type, and diffuse invasion patterns [80]. Primary tumors of different origins were shown to form metastatic lesions of distinct morphologies. Breast carcinoma metastases predominantly produced papillary-type metastases (Figure 5A), while pushing- (Figure 5B) and mixed phenotypes were distinctive for colon and lung carcinoma metastases. A prominent feature of pushing-type metastases was the formation of a multicellular PDGFRβ+ pericyte layer embedded within a thickened vascular basement membrane. By contrast, thickening of the pericyte layer was uncommon in papillary-type metastases and restricted to vessels at the metastatic site.

Metastatic growth was accompanied by an increased pericytic expression of Serpin H1 (an enzyme involved in collagen biosynthesis) and αSMA, which coincided with elevated levels of collagen in the vessel walls [80]. Hence, pericytes possess the capability to differentiate into other cell types and may be viewed as the main source of the connective tissue in human parenchymal BM. In human BCBM xenograft, MDA-MB-231/BR or HER2 overexpressing MCF-7 cells secrete PDGF-BB [136,137]. A high PDGF-BB gradient arising from the tumor attracted the pericytes to move toward tumor cells. This caused the separation of tumor cells from tumor microvessels, stimulated pericyte-to-fibroblast-like transition, the expression of fibroblast-specific protein 1 (FSP-1), and promoted tumor invasion and metastasis [84]. Moreover, it is speculated that pressure exerted from the growing tumor may result in the acquisition of an α-SMA-positive pericytic phenotype with cellular transformation into highly contractile pericytes capable of bracing the expansive force of the tumor [80].

Several findings suggest a critical role of pericytes in the development of BM lesions by directly influencing key steps of tumor progression. In a syngeneic mouse 4T1 mammary carcinoma model of brain metastasis, pericytes were shown to be located adjacent to capillaries within BM lesions [138]. In vitro, MDA-MB-231 triple-negative BC cells preferentially co-localized with pericytes and actively migrated closer towards pericytes than in the direction of EC [138]. Insulin-like growth factor 2 (IGF2) is expressed in and secreted by brain pericytes, and blocking IGF1R with the selective IGFR1 inhibitor picropodophyllin was shown to reduce BC cell proliferation and adhesion and to reduce the size of 4T1 brain metastatic lesions [138]. Enhanced adhesion of tumor cells in pericyte-conditioned medium might be explained by the pericyte-derived secretion of collagen type IV and fibronectin, which are major components of vascular BM in the brain [138,139].

## 11. In Vitro and In Vivo Models in Pericyte Research

### 11.1. Pericytes Co-Culture Models

Pericytes have been predominantly investigated for their role in BBB functions. Several in vitro BBB models have demonstrated the role of pericytes in supporting BBB function [98,99,140,141]. Measuring the transendothelial electrical resistance (TEER) to assess EC barrier tightness has become the standard for determining the success of in vitro models at recapitulating BBB properties characterized by high TEER values and low permeability [142,143]. Models with primary endothelial cells and pericytes isolated from mouse, rat, and porcine brains are commonly used [142]. A rat BBB model with a co-culture of primary brain EC, pericytes, and astrocytes [98,99] demonstrated that these co-culture conditions improved the tightness of the EC barrier, indicating that cellular signals from pericytes and astrocytes may modulate BBB properties [98,99]. The effect of pericytes on brain EC barrier functions was found to be dependent on the differentiation stage and tissue origin of the pericytes [144]. α-SMA-positive, but not negative, pericytes induced higher TEER values in the co-cultures of primary porcine brain EC with pericytes [141], and growth factors such as TGFβ could upregulate α-SMA in pericytes [141,144]. Using a BBB co-culture system of human cells demonstrated that cerebral microvascular EC generated increased TEER when co-cultivated with cerebral astrocytes and brain vascular pericytes (HBVPs). This increase in TEER coincided with the expression of tight junction proteins claudin-5 and ZO-1 in the endothelial layer [140]. Hence, a sophisticated in vitro modeling system must consider the dynamic impact of co-culture conditions on the differentiation potential and functions of pericytes by ideally performing live monitoring of pericyte differentiation in a multi-cellular system equipped with highly sensitive sensors to assess functional changes in real time [65]. The use of primary cells isolated from brain tissue or immortalized cell lines can present limitations due to their suboptimal barrier properties or difficulties in obtaining sufficient quantities of isolated brain primary cells for co-culture [145]. To overcome these obstacles and improve the scalability of human models, BBB models using human pluripotent stem cell (hPSC)-derived cell components have been developed [36,145,146]. Brain pericyte-like cells with robust expression of pericyte markers were obtained through mesodermal and neural crest differentiation of hPSCs [36]. Such cells maintained stable expression of CD13 and PDGFR markers when used in co-cultures [146]. The functionality of hPSC-derived pericytes was found to be comparable to HBVPs [36,146]. Similar results were observed in pericytes derived from patient-derived iPSCs, providing a promising pericyte cellular tool for future disease modeling [146].

Replicating the functions of pericytes for a functional BBB requires culture conditions that reflect the proper cellular and spatial organization of the NVU. Transwell inserts are most commonly used for in vitro models of the BBB [147] for studying cell–cell interactions that affect barrier functions [98,140,141] as well as cancer cell migration across the endothelium [100]. Transwell inserts have a porous filter membrane that separates the well into luminal and abluminal compartments while allowing communication between cells in both compartments. Endothelial cells are typically seeded on the luminal (upper) side of the insert. Pericytes or astrocytes can be seeded on the abluminal (lower) side to permit direct contact with endothelial cells through membrane extensions or on the bottom of the well to make a non-contact model. In both animal and human cell models, contact co-cultures were able to induce higher TEER than with non-contact cultures, suggesting the importance of cell–cell contacts between pericytes and EC for increased barrier functions [98,140]. Transwell assays are also useful in studying the process of cancer cell migration across the BBB. Fluorescently labeled tumor cells seeded on top of an EC monolayer on the luminal side of the insert mimic cancer cells in the vasculature and the interaction with the endothelium [100,148,149]. In addition to improving EC barrier function, the presence of brain pericytes reduced the permeability of the endothelium to lung cancer cells and inhibited cancer cell colony growth on the “luminal” side of the chamber [100]. These studies suggest that brain pericytes may indirectly inhibit cancer cell colonization of the brain. However, their direct role on cancer cells on the abluminal side has not been studied and more research is needed to understand the role of brain pericytes on cancer cells in close proximity in the perivascular space (Figure 6).

### 11.2. Microfluidics—Dynamic In Vitro Modeling of the Brain Microvasculature

Transwell systems are limited by the large fluid-to-cell volume ratio and lack of flow, which is a critical parameter in blood vessels and a major contributor of shear stress in EC [150]. Microfluidic devices can overcome these drawbacks by providing a highly controlled platform capable of better mimicking the in vivo microenvironment [151] (Figure 7). Microfluidic BBB models allow for co-culturing of multiple cell types in a flow-through system with simultaneous high-resolution imaging and real-time monitoring of cellular migratory responses [151]. A microfluidic system comprised of human iPSC-derived EC, human brain pericytes, and astrocytes was shown to exhibit perfusable vasculature with low permeability values that were comparable to the in vivo rat brain [152]. A similar microfluidic co-culture model demonstrated specific BBB properties, including a narrow vessel lumen, high tight junction protein expression with low permeability, and a functional efflux transport system [153]. Other microfluidity systems have been established with human EC differentiated from iPSC in conjunction with primary brain pericytes and astrocytes [154,155] that demonstrate similar in vivo-like barrier functions and are promising screening models to assess pharmacological drugs for improved BBB passage to the brain [153]. These microfluidity models have primarily been created for drug testing but likely are suitable to investigate cancer cell endothelial transmigration in real time, under flow conditions and with much smaller numbers of cancer cells. A point of caution is the presence of αSMA+ pericytes, which may reflect an in vitro-induced pericyte differentiation phenotype that could bias the experimental outcome.

Microfluidic devices are promising BBB model systems that provide the in vivo microenvironment more closely and have the potential to be adapted for high-throughput clinical studies [156]. Microfluidic organ-on-a-chip assays are emerging that study intercellular communication in “brain-on-chip” modeled tissue- and organ-specific cellular compartments and are summarized elsewhere [157]. These multicellular brain-on-chip in vitro models combine capillary flow with multi-cellular brain compartments and allow for the monitoring of cell proliferation and migration in the context of relevant cell connections [158], as well as cell responses from paracrine signaling between compartments [159]. Table 1 summarizes in vitro models for the BBB.

### 11.3. In Vivo Monitoring of Pericytes

The high variability in the temporal and spatial occurrence of hematogenic brain metastasis imposes great challenges for in vivo tracing and monitoring of pericyte functions during the initiation and formation of brain metastases. Two-photon imaging through a chronic skull window has been used for in vivo pericyte imaging by monitoring calcium flux in anesthetized transgenic mice expressing genetically encoded calcium sensors in ensheathing pericytes [160]. A multimodal imaging approach was described to visualize NG2-tdTomato-labeled pericytes in mouse brains. Transcranial two-photon microscopy with a 3D imaging volume of 500 × 500 × 250 μm combined with transcranial epifluorescence time-lapse microscopy allowed the study of pericyte turnover after seizure induction in mice and treatment with PDGF-BB [161]. The fluorescent Nissl dye NeuroTrace 500/525 specifically labels αSMA-negative pericytes in brain capillaries in vivo, thus enabling in vivo imaging of selected pericyte subpopulations devoid of αSMA in mouse brains [67]. The topical dye application caused uptake by PDGFRβ+ pericytes, which remained labeled for up to 3 days allowing for in vivo monitoring to a depth of 400 μm using intracranial two-photon microscopy. However, the requirement for a localized cranial window and the duration of the anesthesia for a pre-defined observational time windows currently limit the use of these methods for metastasis research. The zebrafish emerges as a promising in vivo model to study brain pericytes [162]. Platelet-derived growth factor receptor β *(Pdgfrβ),* notch receptor 3 *(Notch3*), and neural/glial antigen 2 *(Ng2)* depict pericytes in zebrafish. Transgenic lines are suitable for lineage tracing and monitoring pericyte plasticity in transgenic zebrafish, and the short developmental duration and the transparency of the zebrafish larvae allow for in vivo lineage tracing of mural cells during development [163] and make this fish model a promising and powerful in vivo tool to study the role of pericytes in brain metastasis in the future. Brain tissue clearing techniques [164] suitable for brain imaging in adult zebrafish contribute to the 3D visualization of brain cells. A summary of suitable pericyte in vivo models is shown in Table 2.

## 12. Conclusions

A better understanding of the interactions between BC and the NVU as well as the BCBM tumor microenvironment is critical in advancing effective treatments against fatal brain metastases. Newly emerging in vivo molecular imaging and tissue profiling technologies are expected to reveal detailed gene and/or protein expression patterns at a high contextual and spatiotemporal resolution. This will excel our understanding of the diversity and functional roles of pericyte populations in normal and diseased tissues, including the NVU and brain metastases. Putting these spatial tissue data to the test in sophisticated in vitro devices as well as genetic and tumor animal models will contribute major advances to our understanding of BCBM and be instrumental in the development of new and more efficacious treatment options for breast cancer and other cancer patients with brain metastasis.

## Figures and Tables

**Figure 1 cells-11-01263-f001:**
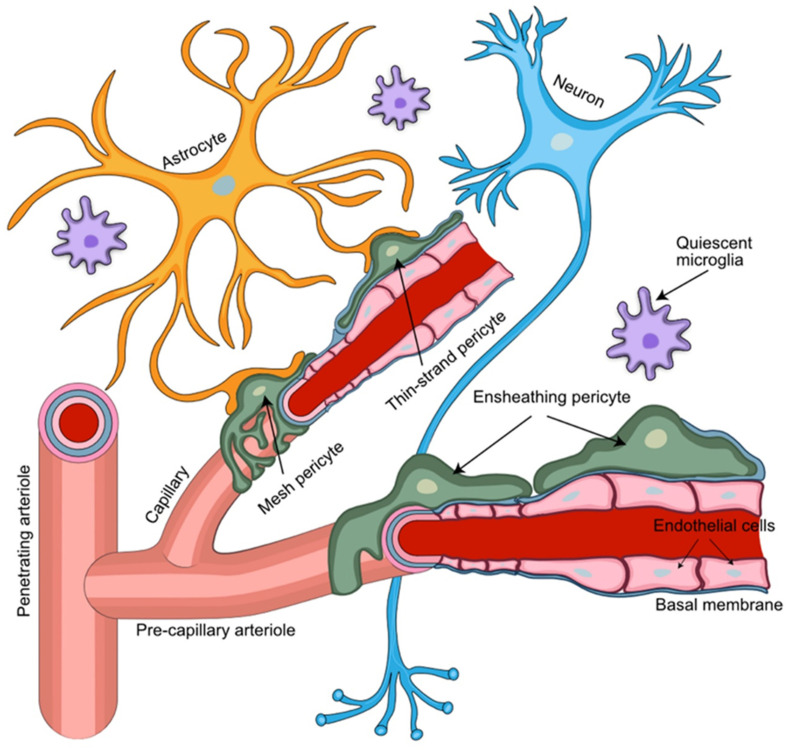
**Perivascular arrangement of pericytes among NVU cellular structures**. Pericytes extend their ramified processes along the brain vasculature, coming into direct contact with the vascular endothelium and astrocytic end-feet. Pericytes embedded in the capillary basement membrane, together with the endothelial cells of the capillary wall and astrocytic end-feet, form the blood–brain barrier. Different types of pericytes are found in distinct locations of the NVU. Ensheathing pericytes occupy mostly pre-capillary arterioles, while mesh and thin-strand pericytes are found mainly at the capillary portion of the brain microvasculature.

**Figure 2 cells-11-01263-f002:**
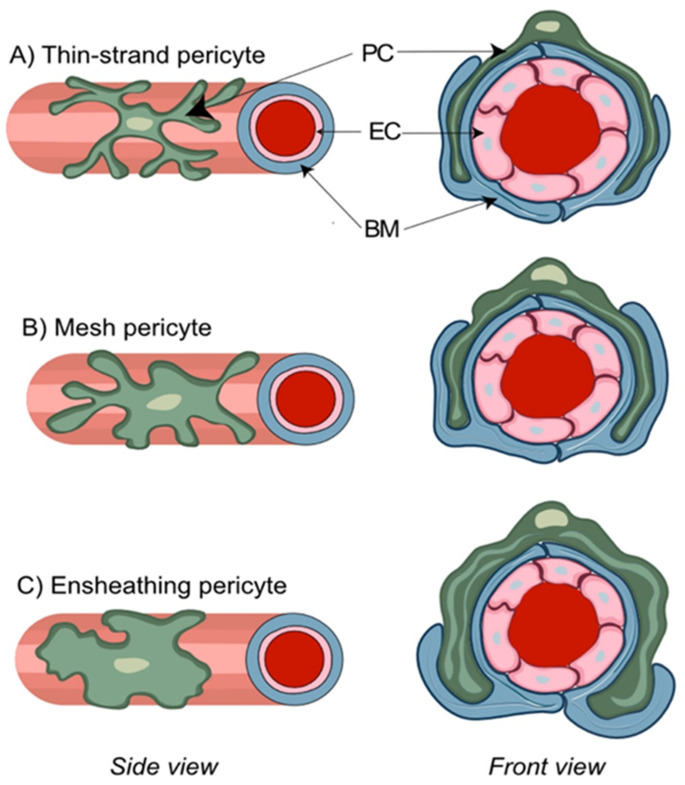
**Morphological features of distinct pericyte (PC) subtypes**. PC are characterized by a typical “bump on a log” appearance arising from an ovoid protruding nucleus and extending cytoplasmic processes that contact the abluminal side of the vascular basal membrane (BM) and run inside of its duplicature. Thin-strand pericytes (**A**) possess relatively long thin cytoplasmic processes with a complex branching pattern that sparsely cover brain capillaries. Mesh pericytes (**B**) have a less complex branching pattern, and their thicker and shorter processes provide more comprehensive coverage of brain capillaries. Ensheathing pericytes (**C**) are characterized by the least complex branching pattern and almost complete coverage of pre-capillary arterioles.

**Figure 3 cells-11-01263-f003:**
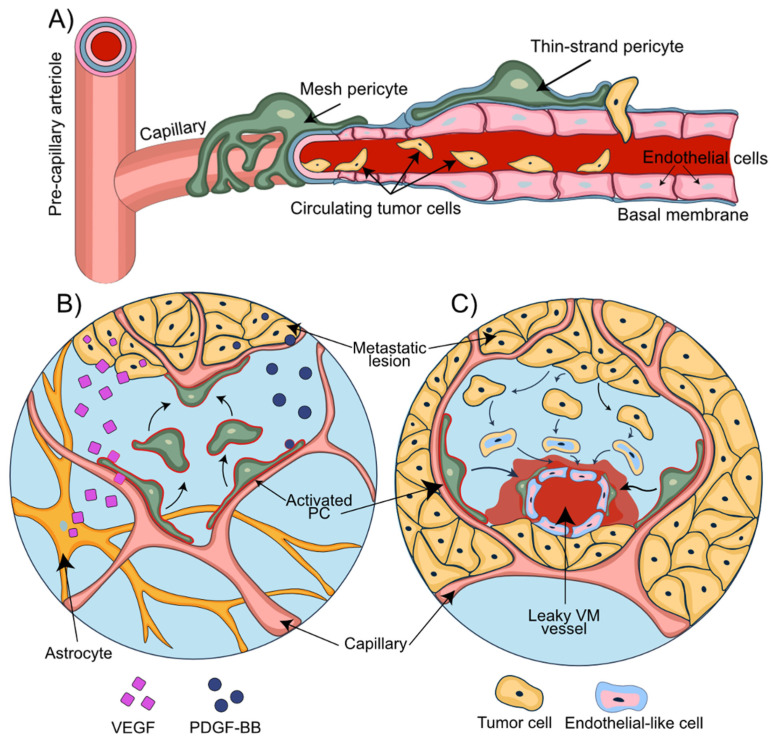
**Pericytes support the disruption of the NVU during metastasis**. (**A**) A consecutive process of rolling, adhesion, and transendothelial migration of circulating tumor cells results in the extravasation of the metastatic breast cancer cells. (**B**) Tumor and astrocyte-secreted VEGF contributes to vessel co-option. PDGF-BB secreted by the metastatic tumor cells mediates displacement of pericytes and their relocation towards the co-opted and newly formed blood vessels. (**C**) Transformation of the tumor cells into endothelial-like cells allows the formation of the vascular mimicry vessels. Pericytes migrate towards these newly formed leaky vascular mimicry vessels and stabilize them.

**Figure 4 cells-11-01263-f004:**
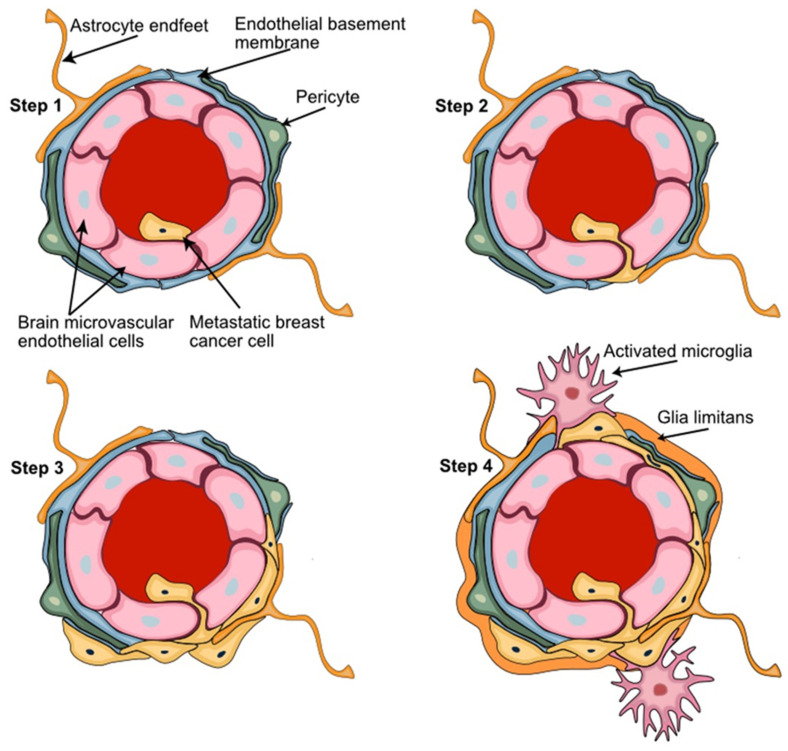
**Key steps of the breast-to-brain metastatic process**. (**Step 1**) Adhesion of the metastatic circulating breast cancer cells to the brain microvascular endothelial cells; (**Step 2**) BBB passage of metastatic cells leaving the circulation by transendothelial migration; (**Step 3**) adaptation and proliferation of the metastatic cells in their new perivascular niche; (**Step 4**) establishment of the tumor microenvironment with metastatic breast cancer cells interacting with the key NVU cellular partner pericytes, astrocytes, and microglia. Astrocyte foot processes associated with the parenchymal basal lamina form the outermost layer of central nervous tissue known as glia limitans.

**Figure 5 cells-11-01263-f005:**
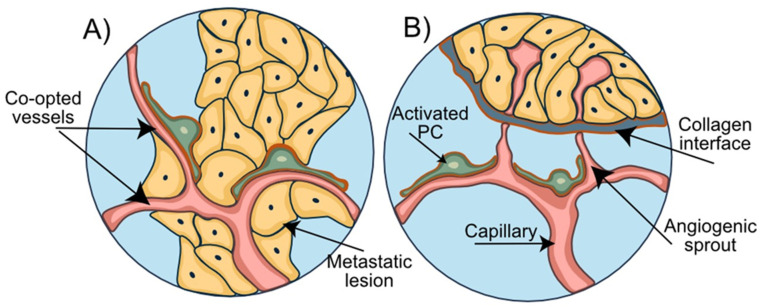
**Invasion patterns of breast-to-brain metastasis**. (**A**) Breast-to-brain metastases are of the papillary type and characterized by the abundant vessel co-option and close direct contacts between the breast cancer cells, blood vessels, and pericytes. (**B**) Brain metastases of the pushing type, originating, for example, from colon cancer, show a distinct collagen layer separating the metastatic lesion from the capillaries and surrounding pericytes.

**Figure 6 cells-11-01263-f006:**
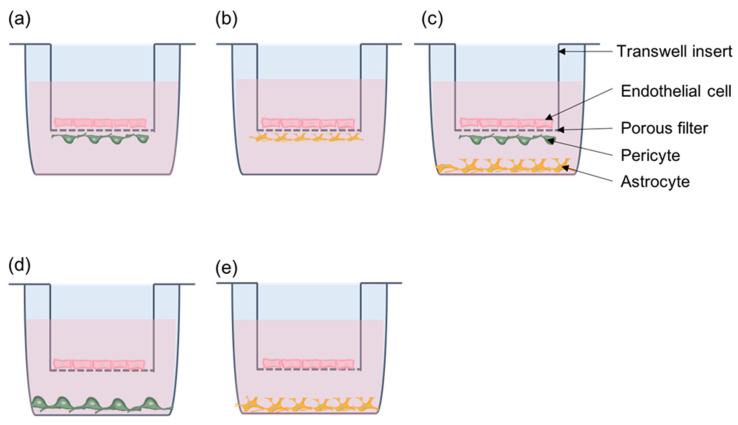
**Transwell system for co-culture**. (**a**) Direct contact model with endothelial cells and pericytes. (**b**) Direct contact model with endothelial cells and astrocytes. (**c**) Tri-cellular culture model with endothelial cells, pericytes, and astrocytes. Endothelial cells and pericytes are in contact. (**d**) Non-contact model with endothelial cells and pericytes. (**e**) Non-contact model with endothelial cells and astrocytes.

**Figure 7 cells-11-01263-f007:**
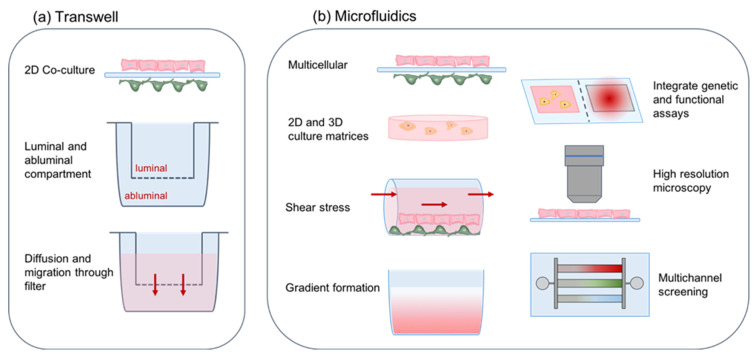
**Properties of BBB in vitro models**. (**a**) Transwell systems utilize a porous filter insert that divides culture wells into luminal and abluminal compartments and allows for diffusion of molecules and cell migration across the filter. (**b**) Microfluidic devices recapitulate a more complex tissue environment in vitro by co-culturing cells in 2D and 3D matrices. Parameters including shear stress and gradient formation are tunable to establish a controlled microenvironment. Microfluidics enable integration of sensing systems and high-resolution microscopy. Multichannel designs allow samples to be screened in parallel, creating compact medium-throughput systems for simulating biological conditions and enabling drug screening. Brain EC are shown in pink on the luminal side, and brain pericytes are marked in green on the abluminal side.

**Table 1 cells-11-01263-t001:** Summary of in vitro BBB models.

Model	Species	Cells in Co-Culture	Reference
Transwell	Rat	Primary cerebral pericytes Primary brain capillary EC Primary cerebral astrocytes	Nakagawa et al., 2009
Transwell	Porcine	Primary brain EC Primary astrocytes	Malina et al., 2009
Transwell	Porcine	Primary brain capillary pericytes Primary brain capillary EC	Thanabalasundaram et al., 2010
Transwell	Porcine	Primary cerebral pericytes Primary brain EC Primary astrocytes	Thomsen et al., 2015
Transwell	Human	Primary brain vascular pericytes Primary cerebral microvascular EC Cerebral astrocytes	Hatherell et al., 2011
Transwell	Human	Primary fetal brain pericytes hPSC-derived brain microvascular EC Differentiated neural progenitor cells	Lippman et al., 2014
Transwell	Human	Primary brain pericytes hPSC-derived pericytes hPSC-derived neural crest stem cells iPSC-derived brain microvascular EC	Stebbins et al., 2019
Microfluidic	Mixed	iPSC-derived human brain microvascular EC Primary rat astrocytes	Wang et al., 2016
Microfluidic	Human	Primary brain pericytes iPSC-derived EC Primary brain astrocytes	Campisi et al., 2018
Microfluidic	Human	Primary placental pericytes Primary brain microvascular EC Primary umbilical vein EC Primary astrocytes	Lee et al., 2019
Microfluidic	Human	Primary brain pericytes iPSC-derived brain microvascular EC Primary astrocytes	Park et al., 2019
Microfluidic	Human	Primary brain pericytes iPSC-derived brain microvascular EC Primary astrocytes	Noorani et al., 2021
Brain-on-chip	Human	Primary brain pericytes, vascular endothelial cells and astrocytes Human neurons differentiated from neuronal stem cells	Maoz et al., 2018
Brain-on-chip	Human	Primary brain pericytes, vascular endothelial cells and astrocytes iPSC-derived cortical neurons and astrocytes	Brown et al., 2016
Brain-on-chip	Human	Different combinations of human brain and NVU cells in review	Saliba et al., 2018

**Table 2 cells-11-01263-t002:** Summary of in vivo models to study pericyte function.

Model	Species	PC Visualization	Reference
Cranial window	Mouse	Transgenic mice; αSMA promoter to label ensheathing pericytes; 2-Photon microscopy	Meza-Resillas et al., 2021
Transcranial imaging	Mouse	Neurotrace™ labeling of pericytes; multimodal optical transcranial imaging	Arango-Lievano et al., 2020
Whole brain imaging	Zebrafish	Transgenic zebrafish; confocal microscopy	Bahrami et al., 2018
Whole mount imaging	Zebrafish	*pdgfrb* promoter transgenic zebrafish; confocal upright fluorescence imaging	Ando et al., 2016
Cranial window	Mouse	αSMA-mCherry transgenic reporter mice; NeuroTrace 500/525 and TO-PRO-3 PC labeling; optogenetic manipulation of PC in rhodopsin transgenic mice	Tong at al., 2021

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
