# Peer review of "Brain Microvascular Pericytes—More than Bystanders in Breast Cancer Brain Metastasis"

_cells, 2022, doi:10.3390/cells11081263_

Round 1
Reviewer 1 Report
The manuscript is very interesting and useful for future research on the brain metastasis process.
Minor concerns regard the style, incorrect syllable synonyms or division (389-390), and many typographical mistakes.
For example:
In the abstract, line 7, and keywords, line 18: please change bold into normal for 'brain'
Line 73: there is a lack of ‘that’
Lines 100 & 231: please extend the abbreviations of AC
Line 211: do you mean precapillary arterioles or capillaries or both?
Line 217: please extend the abbreviations of approx.
Line 222: there is a lack of ‘as’
Line 22: typo for perictyes
Lines 257, 280, 479, 530, etc.: please add a space
Lines 365, 385: probably there is a lack of beta (PDGFβR)
Lines 583-584: are NG2 and CSPG4 two different genes/proteoglycans in zebrafish? I don’t think so.
Could be useful to use PC for pericyte?
Could be more elegant to put figure 3A on 3B,C?
Could be more elegant to add an additional arrow to indicate the porous filter?
Reviewer 2 Report
This review article has summarized the current understanding of the contribution of pericytes in diverse pathologies in the brain. The review gives an interesting scientific perspective on this topic, focusing on known phenotypes of pericytes in breast cancer brain metastasis and discusses the diverse markers for brain pericytes. In addition, the authors review current in-vitro and in-vivo experimental models to study pericyte function. This review addressed a very interesting and important topic. The information gathered in this review is rich and well organized.
Minor Comments:
1)Although the authors mentioned the previous development on BBB-Chip models, they did not include more recent studies on Brain-Chip that recapitulate the entire neurovascular unit (including neurons). This should be discussed in the manuscript.
2)What is missing is a table that summarizes the current in-vitro and in-vivo experimental models to study pericyte function.
